# A toolbox for ablating excitatory and inhibitory synapses

Aida Bareghamyan[1,2], Changfeng Deng[3], Sarah Daoudi[1], Shubhash C Yadav[4], Xiaocen Lu[5], Wei Zhang[5], Robert E Campbell[5,6], Richard H Kramer[4], David M Chenoweth[3], Don B Arnold[1,2,7]*

[1]Department of Biology, Division of Molecular and Computational Biology, University of Southern California, Los Angeles, United States; [2]Neuroscience Graduate Program, University of Southern California, Los Angeles, United States; [3]Department of Chemistry, School of Arts and Sciences, University of Pennsylvania, Philadelphia, United States; [4]Department of Molecular and Cell Biology, University of California, Berkeley, United States; [5]Department of Chemistry, Faculty of Science, University of Alberta, Edmonton, United States; [6]Department of Chemistry, Graduate School of Science, The University of Tokyo, Bunkyo-ku, Tokyo, Japan; [7]Department of Biomedical Engineering, Viterbi School of Engineering, University of Southern California, Los Angeles, United States

*For correspondence:
darnold@usc.edu

## eLife Assessment

This **compelling** study introduces a set of novel genetically encoded tools for the selective and reversible ablation of excitatory and inhibitory synapses. These new tools enable selective and efficient ablation of excitatory synapses, and photoactivatable and chemically inducible methods for inhibitory synapse ablation in specific cell types, providing **valuable** methods for disrupting neural circuits. This approach holds broad potential for investigating the roles of specific synaptic input onto genetically determined cells.

**Abstract** Recombinant optogenetic and chemogenetic proteins are potent tools for manipulating neuronal activity and controlling neural circuit function. However, there are few analogous tools for manipulating the structure of neural circuits. Here, we introduce three rationally designed genetically encoded tools that use E3 ligase-dependent mechanisms to trigger the degradation of synaptic scaffolding proteins, leading to functional ablation of synapses. First, we developed a constitutive excitatory synapse ablator, PFE3, analogous to the inhibitory synapse ablator GFE3. PFE3 targets the RING domain of the E3 ligase Mdm2 and the proteasome-interacting region of Protocadherin 10 to the scaffolding protein PSD-95, leading to efficient ablation of excitatory synapses. In addition, we developed a light-inducible version of GFE3, paGFE3, using a novel photoactivatable complex based on the photocleavable protein PhoCl2c. paGFE3 degrades Gephyrin and ablates inhibitory synapses in response to 400 nm light. Finally, we developed a chemically inducible version of GFE3, chGFE3, which degrades inhibitory synapses when combined with the bio-orthogonal dimerizer HaloTag ligand-trimethoprim. Each tool is specific, reversible, and capable of breaking neural circuits at precise locations.

**eLife digest** Within the brain are circuits of neurons that communicate with one another via junctions known as synapses. The pre-synaptic neuron, which sends the signal, releases proteins known as neurotransmitters into the synapse which then bind to receptors on the receiving, or postsynaptic, neuron.

If the receptors at the synapse are excitatory, they increase the chances of the postsynaptic neuron 'firing' and propagating the signal. In contrast, if the receptors are inhibitory, this reduces the likelihood of the neuron firing, dampening communication across the circuit.

In a previous study, a tool known as GFE3 was developed that can specifically eliminate inhibitory synapses by binding to scaffolding proteins called gephyrins that anchor inhibitory receptors in place. Attached to GFE3 is an enzyme that triggers gephyrin degradation, leading to the dismantling of the synapse and blocking of the inhibitory signal. The tool has been used to study various behaviors, including how mice control their rhythmic whisker movements and vocalization patterns.

Bareghamyan et al. – who are part of the research group that carried out the previous work – have now refined GFE3 so it can be activated more precisely and controllably. The team produced two new versions: one that is activated by light, and another that is activated by a cell-permeable chemical compound known as trimethoprim-Halotag ligand.

In addition, Bareghamyan et al. engineered a new tool called PFE3 which is designed to eliminate excitatory synapses. PFE3 targets PSD-95, the predominant scaffold protein found in excitatory synapses. It also contains two enzymes that work together to degrade PSD-95, leading to a loss of excitatory receptors. Further experiments showed that PFE3 efficiently reduced the number of excitatory synapses in rat neurons cultured in a dish as well as in neurons found in the retinas of mice.

The synapse ablators developed by Bareghamyan et al. offer a fast, efficient and reversible approach for eliminating both excitatory and inhibitory synapses. These tools will make it easier for neuroscientists to silence specific postsynaptic neurons and expand the toolkit available for manipulating and studying neuronal circuits.

## Introduction

Neural circuits, groups of neurons connected by synapses, are the basic units of computation in the brain and are fundamental to understanding its function and pathology. Although there are many genetically encoded tools for modulating neuronal function, existing tools for eliminating synapses are inadequate for providing a clear understanding of neural circuits. Previously, we developed GFE3, a protein that can ablate inhibitory synapses efficiently, rapidly, and without toxicity by targeting an E3 ligase to the scaffolding protein Gephyrin (*Gross et al., 2016*). GFE3 consists of GPHN.FingR (**F**ibronectin **in**trabody **g**enerated by m**R**NA display), a recombinant, antibody-like protein, which binds Gephyrin at inhibitory synapses (*Prior et al., 1992*; *Gross et al., 2013*) fused to the RING domain of the E3 ligase XIAP (RING$_{XIAP}$) (*Wilkinson et al., 2008*). GFE3 mediates the ubiquitination of Gephyrin and disassembly of inhibitory synapses. Importantly, it specifically targets postsynaptic sites containing GABA$_A$ receptors and can be expressed in genetically determined cell types. Thus, GFE3 can manipulate circuits in a manner that is not possible with optogenetics (*Yizhar et al., 2011*), DREADDs (*Urban and Roth, 2015*), and modulators of neurotransmitter release (*Liu et al., 2019*), which do not target specific postsynaptic receptors, or with traditional pharmacological approaches that target specific receptors but do not allow for cell-type specificity. GFE3 has been used to probe the contribution of inhibitory inputs to LTP in hippocampal circuits (*Davenport et al., 2021*), recurrent loops in the oscillator that drives rhythmic whisking (*Golomb et al., 2022*; *Takatoh et al., 2022*), and the temporal coordination of vocalization and inspiration (*Park et al., 2024*). However, no tool analogous to GFE3 that ablates excitatory synapses currently exists. Furthermore, because GFE3 is constitutively active, its temporal and spatial resolution is limited.

In this study, we generated three tools for ablating synapses based on GFE3: (1) PFE3, an excitatory synapse ablator; (2) paGFE3, a photoactivatable version of GFE3; (3) chGFE3, a chemically activated version of GFE3. We show that the expression of PFE3 in cultured neurons causes the loss of PSD-95 puncta and excitatory synapses, which is reversible. In vivo, its expression leads to the functional loss of excitatory synaptic transmission. We generated paGFE3 by incorporating GPHN.FingR

and the $RING_{XIAP}$ into a novel photoactivatable complex based on the photocleavable protein PhoCl2c (*Lu et al., 2021*; *Zhang et al., 2017*). paGE3 is activated by 400 nm light and causes ablation of inhibitory synapses within 5 hr of exposure that is subsequently reversible. In the absence of 400 nm light, paGFE3 has no background activity. In addition, the expression of paGFE3 labels inhibitory synapses, allowing their size and location to be monitored before and after ablation. chGFE3, a chemogenetic version of GFE3 analogous to paGFE3, mediates efficient, reversible degradation of labeled synapses when a cell-permeant chemical is added.

## Results

### Degrading exogenous PSD-95 through ubiquitination

Initially, to generate an excitatory synapse ablator, we fused $RING_{XIAP}$ to PSD-95.FingR, which binds at high affinity and specificity to the scaffolding protein PSD-95, which is found at excitatory synapses (*Hunt et al., 1996*). However, PSD-95.FingR-$RING_{XIAP}$ did not efficiently ablate PSD-95 (data not shown). As an alternative, we fused PSD-95.FingR to the RING domain of the E3 ligase Mdm2 ($RING_{Mdm2}$), which is necessary for PSD-95 degradation and excitatory synapse ablation during homeostatic plasticity (*Colledge et al., 2003*). To determine whether PSD-95.FingR-$RING_{Mdm2}$ degrades PSD-95, we examined how co-expression with PSD-95.FingR-$RING_{Mdm2}$ affected the expression of exogenous PSD-95-myc in COS7 cells. Cells transfected with PSD-95 alone had a PSD-95-myc expression level of 203 ± 32 au, as measured by western blot, while the expression level of PSD-95 in cells co-expressing PSD-95-myc and PSD-95.FingR-$RING_{Mdm2}$ reduced to 107 ± 12 au, a ~50% decrease ($n$ = 5, p < 0.05, ANOVA with multiple comparisons). To determine whether this reduction in PSD-95 was due to ubiquitination, we tried a third condition, where the ubiquitination inhibitor TAK243 (*Majeed et al., 2022*) was added to cells co-expressing PSD-95-myc and PSD-95.FingR-$RING_{Mdm2}$. In this case, PSD-95 expression levels increased to 239 ± 31 au, a ~20% increase relative to cells expressing PSD-95-myc alone, which was not significant ($n$ = 5, p > 0.6, ANOVA with multiple comparisons). Furthermore, cells co-expressing PSD-95-myc and PSD-95.FingR-$RING_{Mdm2}$ without TAK243 had ~55% less PSD-95 vs. those with TAK243, a significant difference (p < 0.05, ANOVA with multiple comparisons, *Figure 1A, B*, *Figure 1—figure supplement 1*).

To test whether TAK243 was blocking ubiquitination, we compared western blot staining of lysates from COS7 cells expressing Ubiquitin-HA with and without TAK243. Staining with anti-HA showed a distinctive laddering pattern in the lane corresponding to cells expressing Ubiquitin-HA without TAK243 consistent with ubiquitination, whereas the lanes corresponding to cells expressing Ubiquitin-HA with TAK243 and a control lane with lysate from untransfected cells showed no staining (*Figure 1—figure supplement 1*), confirming that TAK243 blocks ubiquitination. Together, our results are consistent with PSD-95.FingR-$RING_{Mdm2}$ degrading exogenously expressed PSD-95 through ubiquitination in COS7 cells.

### Degradation of endogenous PSD-95 in neurons

To test whether PSD-95.FingR-$RING_{Mdm2}$ can degrade endogenous neuronal PSD-95, we co-transfected doxycycline (Dox)-inducible TRE-PSD-95.FingR-HA-$RING_{Mdm2}$ and transcriptionally regulated PSD-95.FingR-tagRFP in 14 DIV (days in vitro) cultures of rat cortical neurons. Note that PSD-95. FingR-tagRFP efficiently labels endogenous PSD-95, allowing its spatial distribution to be mapped in real time in living cells (*Gross et al., 2013*). Furthermore, transcriptional regulation matches the expression level of PSD-95.FingR with that of endogenous PSD-95, facilitating labeling with very low background. Following incubation for 4 days, we imaged the neurons for PSD-95.FingR-tagRFP and subsequently induced the expression of PSD-95.FingR-HA-$RING_{Mdm2}$ with Dox. After 48 hr, we reimaged the neurons for PSD-95.FingR-tagRFP and then fixed and stained them with anti-PSD-95 and anti-HA. We found that PSD-95.FingR-tagRFP labeling was reduced by 85% (p = 0.002, Wilcoxon, $n$ = 9 cells, 3 independent experiments, *Figure 1C, D*), consistent with efficient ablation of endogenous PSD-95. However, when we counted the number of puncta labeled with PSD-95.FingR at T0 and compared that to the number of puncta labeled with immunostaining of endogenous PSD-95 in the same cells at the end of the experiment, it showed a reduction of only 52% (*Figure 1E, F*, p < 0.0001, Mann–Whitney, $n$ = 9 cells, 3 independent experiments). A strategy for improving this relatively low ablation rate might be provided by the results of experiments where the expression of MEF2 was

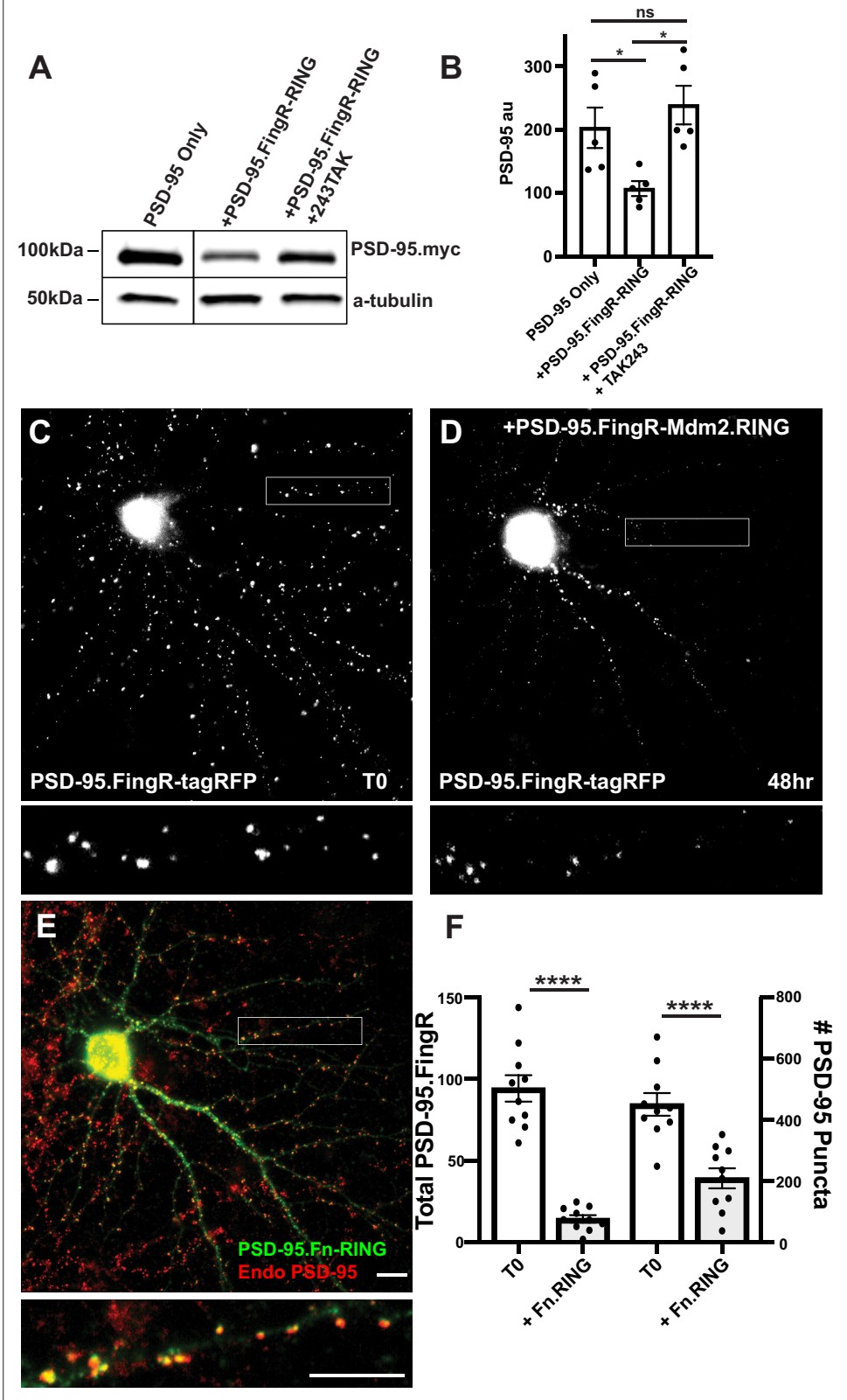

**Figure 1.** Mdm2.RING ubiquitinates PSD-95. (**A**) Comparison of cell lysate from COS7 cells transfected with PSD-95-myc alone, PSD-95-myc + Dox-induced TRE-PSD-95.FingR-RING$_{MDM2}$, and PSD-95-myc + TRE-PSD-95. FingR-RING $_{MDM2}$ + TAK243, a ubiquitination inhibitor. Cells expressing TRE-PSD-95.FingR-RING$_{MDM2}$ were treated with 1 µg/ml doxycycline for 4 hr to induce expression of PSD-95.FingR-RING and showed a reduction in PSD-95-

*Figure 1 continued on next page*

*Figure 1 continued*

myc. Cells treated with 20 μM TAK243 showed no apparent reduction in PSD-95-myc. (**B**) Quantitation showed a significant reduction in PSD-95 expression when co-expressed with PSD-95.FingR-RING$_{MDM2}$ in COS7 cells, but not when PSD-95 is expressed alone or when PSD-95 and PSD-95.Fn-RING $_{MDM2}$ are co-expressed with 20 μM TAK243. *p < 0.05, ANOVA with multiple comparisons. ns, p > 0.05. (**C**) Cultured cortical neuron expressing transcriptionally regulated PSD-95.FingR-tagRFP before induction of PSD-95.FingR-HA-RING$_{Mdm2}$ expression with Dox. (**D**) Same neuron as in (**C**) after induction of PSD-95.FingR-HA-RING$_{MDM2}$ expression with Dox shows a reduction in PSD-95.FingR-tagRFP labeling. (**E**) Immunostaining of the neuron in (**D**) for PSD-95.FingR-HA-RING$_{MDM2}$ (green) and endogenous PSD-95 (red). (**F**) Quantification of the number of the total amount of PSD-95 labeled by PSD-95. FingR-tagRFP before and after expression of PSD-95.FingR-HA-RING$_{MDM2}$ (left side). The # of puncta labeled with PSD-95.FingR-tagRFP at T0 was compared to the number of puncta immunostained for endogenous PSD-95 following expression of PSD-95.FingR-HA-RING$_{MDM2}$ (right side). ****p < 0.0001, Mann–Whitney. Error bars represent ± SEM. Scale bars: 5 μm.

The online version of this article includes the following source data and figure supplement(s) for figure 1:

**Source data 1.** Source data for *Figure 1A*.

**Source data 2.** Source data for *Figure 1A*.

**Source data 3.** Numerical source data for graphs in *Figure 1B, F*.

**Figure supplement 1.** TAK243 blocks ubiquitination and degradation of PSD-95 by Mdm2.RING.

**Figure supplement 1—source data 1.** Annotated western blots for ubiquitin and tubulin.

**Figure supplement 1—source data 2.** Original western blots for ubiquitin and tubulin.

---

found to cause the elimination of excitatory synapses (*Tsai et al., 2012*). In those experiments, efficient synapse elimination was found to require a combination of ubiquitination mediated by Mdm2 and interaction with the proteasome, which was mediated by Protocadherin 10 (PCDH 10). PCDH 10 is a $Ca^{2+}$-dependent cell adhesion protein that binds to both PSD-95 and the proteasome via its proteasome-interacting region (PIR) (*Tsai et al., 2012*). Therefore, we reasoned that adding the PIR domain to the PSD-95.FingR-RING$_{Mdm2}$ complex might increase the efficiency with which PSD-95 is ablated.

## Optimization of degradation using the PIR

We generated a new protein, PSD-95.FingR-RING$_{Mdm2}$-PIR, which we called PFE3. To test PFE3, we co-transfected TRE-PFE3-HA with PSD-95.FingR-tagRFP in dissociated cultures of rat cortical neurons. Following 4 days of incubation, we imaged the PSD-95.FingR-tagRFP and induced the expression of PFE3-HA with Dox. The neurons were imaged 48 hr after induction of TRE-PFE3 and subsequently fixed and immunostained for endogenous PSD-95 and HA (*Figure 2A–C*). By comparing images at T0 and 48 hr, we found that the expression of PFE3 reduced the labeling of PSD-95.FingR-tagRFP by 65%, a significant difference (*Figure 2A, B, D*, n = 14 cells, 3 distinct experiments, p < 0.0001, Mann–Whitney). When we checked these results by counting PSD-95 puncta labeled with PSD-95.FingR-tagRFP at T0 and immunocytochemistry against endogenous PSD-95 at 48 hr (*Figure 2C*), we found that Dox-inducible PFE3 reduced the number of endogenous PSD-95 puncta by 73% (*Figure 2D*, p < 0.0001, Mann–Whitney). Thus, PFE3 consistently and efficiently reduced both total labeling of PSD-95 by the PSD-95.FingR, and the number of endogenous PSD-95 puncta. As a control, we induced RandE3 (Random.FingR-RING$_{Mdm2}$-PIR), which contained a fibronectin scaffold with a random binding pocket instead of PSD-95.FingR (*Gross et al., 2016*). The expression of RandE3 did not significantly affect PSD-95.FingR labeling (an increase of 30%, p > 0.1, Mann–Whitney, n = 8, 3 independent experiments) or the number of endogenous PSD-95 puncta (an increase of 2%, p > 0.9, Mann–Whitney, n = 8, 3 independent experiments, *Figure 2—figure supplement 1A–D*).

To further characterize the effect of PFE3 expression, we determined how it affected the expression of AMPA receptors (AMPARs), which mediate excitatory synaptic transmission (*Gouaux, 2004*). We immunostained non-transfected cultured cortical neurons for PSD-95 and GluA1, a subunit of the AMPAR, and found that 89 ± 1% of PSD-95 puncta were positive for GluA1 (*Figure 2E, H*, 7570 synapses, 11 cells, 3 experiments). Neurons co-transfected with the PSD-95.FingR and TRE-RandE3-HA and induced with Dox for 48 hr showed no significant change in the percentage of PSD-95-positive synapses labeled with GluA1 compared to control neurons (*Figure 2F, H*, 87 ± 2%, 3131

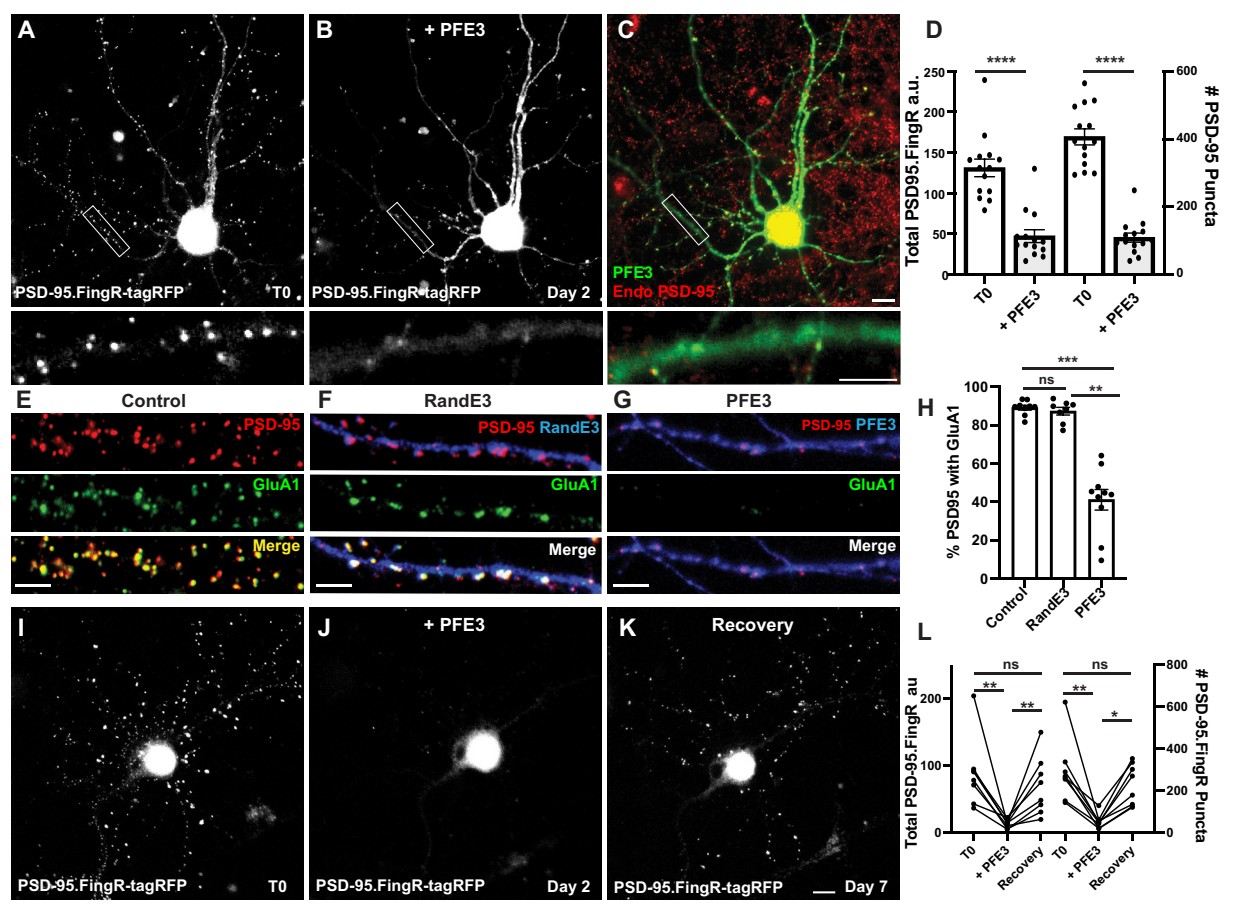

**Figure 2.** PFE3 reversibly ablates PSD-95 in neurons. (**A**) Cultured cortical neuron expressing PSD-95.FingR-tagRFP before induction of PFE3. Closeup of the boxed area shown below. (**B**) Same neuron as in (A) after expression of PFE3 for 48 hr shows a dramatic reduction in PSD-95.FingR labeling. Closeup of the boxed area shown below confirms the lack of punctate labeling by PSD-95.FingR-tagRFP. (**C**) Neuron in B immunostained for PFE3 (green) and endogenous PSD-95 (red) confirms the lack of punctate labeling by PSD-95.FingR. Closeup of the boxed area shown below. (**D**) Quantifications of the total amount of PSD-95.FingR labeling and the # of PSD-95 puncta (T0, # of puncta labeled with PSD95.FingR; +PFE3, # of puncta labeled with immunocytochemistry) show significant reductions following expression of PFE3. ****p < 0.0001, Mann–Whitney. (**E–G**) Cultured cortical neurons immunostained for endogenous PSD-95 in red and GluA1 in green. (**E**) Untransfected. (**F**) Following RandE3 (blue) expression for 48 hr, GluA1 and PSD-95 staining is intact. (**G**) Following PFE3 expression (blue) for 48 hr, GluA1 and PSD-95 expression is markedly diminished. (**H**) Quantification of the percentage of PSD-95 puncta positive for GluA1 staining. ns p > 0.05, ***p < 0.001, **p < 0.01, Kruskal–Wallis multiple comparisons. (**I**) Cultured neuron expressing PSD-95.FingR-tagRFP. (**J**) Same neuron as in (I) 48 hr after induction of PFE3 expression shows a dramatic reduction in labeling with PSD-95. FingR-tagRFP. (**K**) Same neuron as in (I), (J) showing recovery of synapses 5 days after removal of PFE3. (**L**) Quantification of the total amount of PSD-95. FingR-tagRFP labeling and the # of PSD-95.FingR-tagRFP-labeled puncta. **p < 0.01, *p < 0.05, ns p > 0.05, Kruskal–Wallis multiple comparisons. Scale bars: 5 µm.

The online version of this article includes the following source data and figure supplement(s) for figure 2:

**Source data 1.** Numerical source data for graphs in *Figure 2D, H, L*.

**Figure supplement 1.** Expression of RandE3 does not significantly affect PSD-95 expression.

**Figure supplement 1—source data 1.** Numerical source data for graphs in *Figure 2—figure supplement 1D, H, L*.

synapses, 9 cells, 2 experiments, p > 0.05, Kruskal–Wallis multiple comparisons). Quantification of GluA1 in PFE3-expressing cells showed that only 41 ± 5% (1341 synapses, 10 cells, 2 experiments) of the remaining PSD-95 puncta also contained GluA1, a significant decrease from untransfected cells (*Figure 2G, H*, p < 0.001, Kruskal–Wallis multiple comparisons) and RandE3-expressing cells (p < 0.01, Kruskal–Wallis multiple comparisons). Based on our result that PFE3 decreases PSD-95 puncta by 73% (*Figure 2D*) and that only 41% of the remaining puncta are positive for GluA1, we estimate that PFE3 decreases the amount of GluA1 by approximately 90%. Thus, our experiments are consistent with PFE3 mediating a significant reduction in both PSD-95 and GluA1. Furthermore, because AMPARs

are the predominant ionotropic glutamate receptor responsible for excitatory transmission (*Gouaux, 2004*), our results suggest that PFE3 reduces functional excitatory connectivity.

## Degradation of PSD-95 with PFE3 is reversible

To test whether the ablation of excitatory synapses by PFE3 is reversible, we used a previous paradigm for testing GFE3. There, we found that because GFE3 degrades itself, it is sufficient to merely cease the induction of GFE3 expression to reverse its effect (*Gross et al., 2016*). Accordingly, we initially transfected TRE-PFE3-P2A-GFP and PSD-95.FingR-tagRFP and incubated for 4 days. We then induced PFE3 expression with Dox for 48 hr and then replaced the medium with Dox-free medium for an additional 5 days to allow excitatory synapses to recover. Cells were imaged immediately before adding Dox, immediately after removing Dox, and after 5 days of incubation in Dox-free medium, followed immediately by fixation and immunostaining. We found that, as in previous experiments, the induction of PFE3 reduced PSD-95.FingR-tagRFP labeling (−81 ± 5%, *Figure 2I, J, L*, $p < 0.01$, Friedman multiple comparison) and PSD-95.FingR puncta (−80 ± 5%, $p < 0.01$, Friedman multiple comparison, $n = 8$ cells, 4 experiments). Five days following removal of Dox, PSD-95.FingR labeling increased by 450 ± 100%, which was significant (*Figure 2J–L*, $p < 0.01$, Friedman multiple comparison) and the number of PSD-95.FingR puncta increased by 400 ± 65% ($p < 0.05$, Friedman multiple comparison). PSD-95.FingR labeling was not significantly different before adding Dox vs. after its removal (*Figure 2L*, −10 ± 20%, $p > 0.5$, Friedman multiple comparison). Similarly, the number of PSD-95.FingR puncta did not show a significant change before induction of PFE3 vs. after its removal (−11 ± 18%, $p > 0.9$, Friedman multiple comparison). Furthermore, immunocytochemistry of neurons following incubation with Dox-free media showed that endogenous PSD-95 is found on dendritic spines following synapse regrowth (*Figure 2—figure supplement 1J*). Finally, PSD-95.FingR-tagRFP puncta showed a similar distribution overall before and after ablation (*Figure 2—figure supplement 1K*). These results are consistent with PSD-95 degradation by PFE3 being reversible.

In a control experiment with RandE3, neurons showed no significant change in the total PSD-95. FingR labeling (+8 ± 6%, $p > 0.9$, Friedman test multiple comparison, $n = 13$, 2 experiments) or the number of PSD-95.FingR puncta (+8 ± 6%, $p > 0.9$, Friedman test multiple comparison) before and after induction of RandE3 for 48 hr (*Figure 2—figure supplement 1E–H*). Five days after the removal of Dox, both the total PSD-95.FingR labeling and the number of PSD-95.FingR puncta showed a nonsignificant decrease when compared with immediately before Dox addition (*Figure 2—figure supplement 1F–H*, −20 ± 6%, $p > 0.05$, and −15 ± 5%, $p > 0.2$, respectively, Friedman multiple comparison). Comparing the PSD-95.FingR before induction of RandE3 at T0 and 5 days after removal of Dox, we found nonsignificant changes in the total amount of PSD-95.FingR labeling (*Figure 2—figure supplement 1E, G, H*, −14 ± 7%, $p > 0.1$, Friedman multiple comparison) and the number of PSD-95.FingR puncta (−10 ± 6%, $p > 0.7$, Friedman multiple comparison). Immunocytochemistry of the cells 5 days after removal of Dox confirms the labeling by PSD-95.FingR (*Figure 2—figure supplement 1I, J*). Thus, PSD-95 degradation only occurs when the RING$_{Mdm2}$-PIR domains are targeted by PSD-95.FingR. Finally, cotransfection with PSD-95.FingR-tagRFP and TRE-PFE3-P2A-GFP without exposure to Dox results in labeling of PSD-95.FingR-tagRFP that is not significantly different from labeling in cultures transfected with PSD-95.FingR-tagRFP alone, both in terms of total PSD-95.FingR-tagRFP labeling ($p > 0.2$, Mann–Whitney) and the number of labeled puncta ($p > 0.4$, Mann–Whitney, *Figure 2—figure supplement 1L*).

## PFE3 expression blocks excitatory synaptic transmission

To test PFE3 in vivo, we examined the physiological effects of expressing it in the retinas of mice. We crossed a CCK-Cre mouse, which expresses Cre in type 6 cone bipolar cells (*Chhatwal et al., 2007*), with an Ai27 mouse expressing Channelrhodopsin 2-tdTomato (ChR2-tdTomato) in Cre-expressing cells (*Madisen et al., 2012*) to generate mice expressing ChR2-tdTomato in type 6 cone bipolar cells. These mice were injected intravitreally with AAVs encoding either CAG-PFE3-P2A-GFP or CAG-RandE3-P2A-GFP. After 3–4 weeks, retinas were isolated and mounted in a chamber for patch clamp recording as previously described (*Jones et al., 2012*). Retinas were perfused in ACSF (Artificial Cerebrospinal Fluid) bubbled with 95% $O_2$/5% $CO_2$. ACET (1 μM) and L-AP4 (10 μM) were added to block synaptic transmission from photoreceptors to Off and On bipolar cells, respectively, to eliminate their natural response to light. Infected cells were identified by GFP expression (*Figure 3A*). GFP-positive

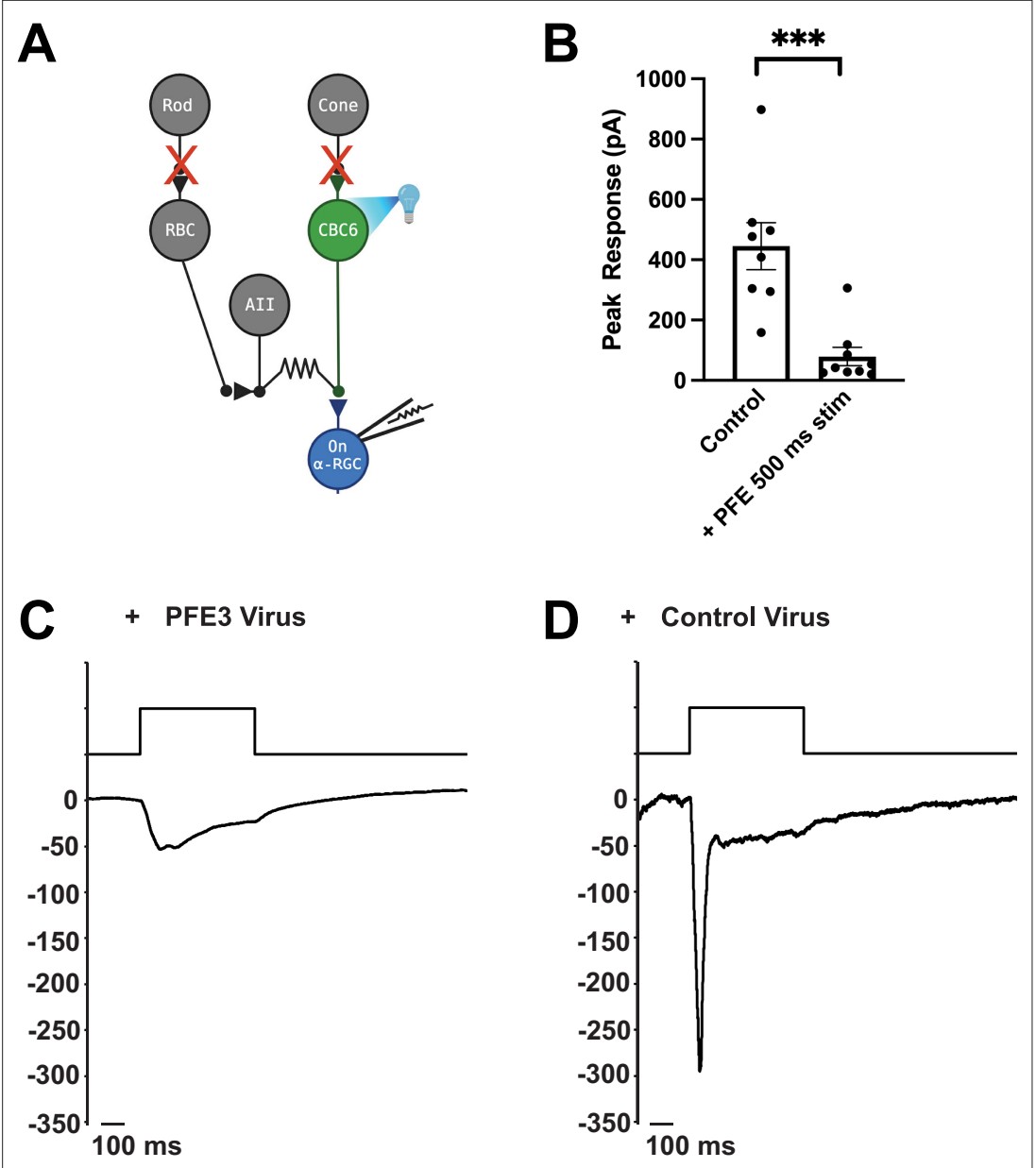

**Figure 3.** Expression of PFE3 reduces synaptic transmission in vivo. (**A**) Schematic depiction of the retinal circuit used to assess AMPA receptor function. Channelrhodopsin-2 is expressed in type 6 cone bipolar cells (CBC6), presynaptic to the On α-retinal ganglion cell (α-RGC). The synaptic output of photoreceptors is pharmacologically blocked. (**B**) Population data comparing peak EPSC amplitude for retinas from mice infected with PFE3 and control viruses. Responses were generated with a 495-nm, 500-ms flash of light. Symbols are values for individual cells. ***p < 0.001, Mann–Whitney. Error bars represent ± SEM. Sample whole cell patch clamp recording from an On α-RGC infected with the PFE3 virus (**C**) or a RandE3 control virus (**D**).

The online version of this article includes the following source data for figure 3:

**Source data 1.** Numerical source data for plots and graphs in *Figure 3B–D*.

On Alpha retinal ganglion cells (α-RGCs) were identified by their large cell bodies, and their identity was confirmed by dye filling. Full-field 495 nm light (0.9 mW/cm$^2$) was delivered to the retina for 500 ms to trigger EPSCs (*Figure 3A*). Synaptic currents were evoked by optogenetic stimulation of type 6 bipolar cells, which provide approximately 70% of the total synaptic input to On α-RGCs (*Schwartz et al., 2012*). At a holding potential of –60 mV, these synaptic currents are almost purely AMPAergic, as the AMPAR antagonist DNQX blocks the response (*Tien et al., 2017*). Compared to RGCs infected with the control virus, RGCs infected with the PFE3 virus had EPSCs with >80% reduction in amplitude,

79 ± 30 pA (*n* = 9 cells, 5 mice) vs. 445 ± 78 pA (*n* = 8 cells, 3 mice), a significant difference (p < 0.001, Mann–Whitney, *Figure 3B–D*), which is consistent with a reduction in postsynaptic receptor density. Overall, our results are consistent with PFE3 expression leading to the degradation of PSD-95, which, in turn, reduces the number of AMPA receptors at excitatory synapses, causing a reduction in synaptic transmission.

## Photoactivatable degradation of inhibitory synapses

To generate a photoactivatable version of GFE3 to ablate inhibitory synapses, we first used existing photodimerization systems, such as the Cryptochrome (Cry2) system, but we found them to have substantial background activity leading to loss of inhibitory synapses even in the dark (*Figure 5—figure supplement 1A*). To overcome this deficiency, we developed a new photoactivatable complex based

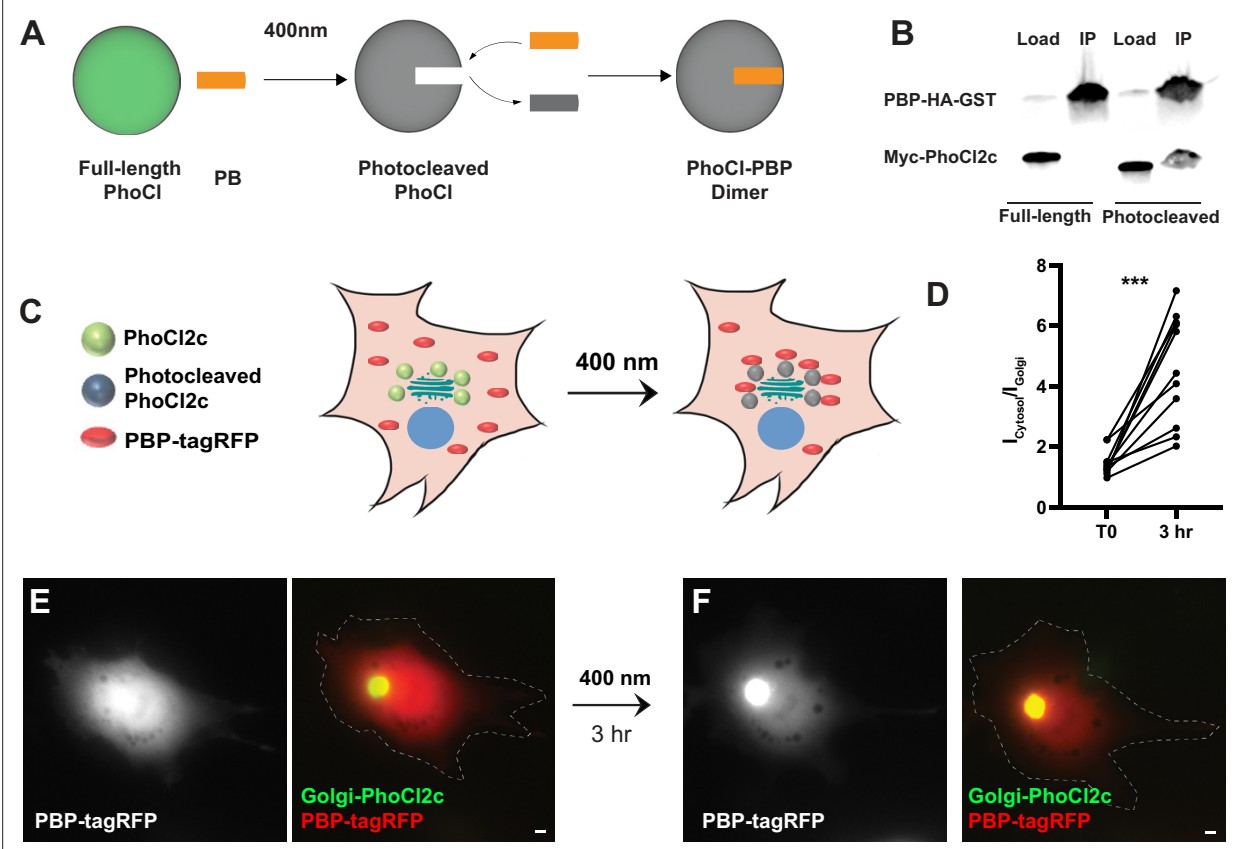

**Figure 4.** Testing of PhLIC, a photoactivatable dimer. (**A**) Schematic of PhLIC (**Ph**oCl-based **L**ight-**I**nducible **C**omplex). Full-length PhoCl is photocleaved with 400 nm light, exposing an epitope recognized by the PhoCl-binding peptide (PBP), leading to dimerization of photocleaved PhoCl and PBP. (**B**) COS7 lysate expressing PBP-HA-GST was incubated with purified full-length or photocleaved Myc-PhoCl2c. PBP-HA-GST pulls down photocleaved but not full-length PhoCl2c. (**C**) Schematic of Golgi-targeting assay. COS7 cell expressing Golgi-targeted PhoCl2c in green and PBP-tagRFP in red before (left) and 3 hr after illumination with 400 nm light for 10 s every 30 s for 3 min (right). (**D**) Quantification of the ratio of the total fluorescence associated with PBP-tagRFP localized in the cytosol ($I_{cytosol}$) vs. at the Golgi ($I_{Golgi}$) before and 3 hr after photocleaving PhoCl2c. ***p < 0.001, Wilcoxon. (**E**) COS7 cell before photoactivation showing PBP-tagRFP (left) and a merge (right) of PBP-tagRFP (red) and PhoCl2c-GTS (green). (**F**) Same COS7 cell as in (**E**) following photoactivation with 400 nm light for 1 min showing PBP-tagRFP (left) and a merge (right) of PBP-tagRFP (red) and PhoCl2c (green). Scale bar represents 5 μm.

The online version of this article includes the following source data and figure supplement(s) for figure 4:

**Source data 1.** Source data for *Figure 4B*.

**Source data 2.** Source data for *Figure 4b*.

**Source data 3.** Numerical source data for graph in *Figure 4D*.

**Figure supplement 1.** PhLIC does not activate with ambient light.

**Figure supplement 1—source data 1.** Numerical source data for graphs in *Figure 4—figure supplement 1E, I*.

on the intrinsically photocleavable protein PhoCl. PhoCl undergoes cleavage when exposed to violet (~400 nm) light, creating two peptides, the C-terminal 10 amino acids, including the chromophore, and the remainder of the protein, referred to as the 'empty barrel' (*Zhang et al., 2017*). Following photocleavage, a novel epitope is exposed on the empty barrel. Based on this property of PhoCl, we developed a photoactivatable complex, PhLIC (**Ph**oCl-based **L**ight-**I**nducible **C**omplex), consisting of the empty barrel and a 10-mer peptide called PhoCl-binding peptide (PBP) that binds to the empty barrel but not to full-length PhoCl (*Figure 4A*). Because PhoCl does not undergo cleavage in the dark or ambient light (*Zhang et al., 2017*), there should be no background formation of PhLIC. To generate PBP, we used the in vitro directed evolution technique, mRNA display, which we have used to create binders to synaptic proteins (*Gross et al., 2013*; *Mora et al., 2013*; *Roberts and Szostak, 1997*; *Figure 4—figure supplement 1A*). We first constructed a library of ~$10^{14}$ semi-random 10-mers, where each amino acid was biased toward the corresponding amino acid in the original C-terminus of PhoCl. Following the first selection, which lasted six rounds, we found PBPs bound to cleaved PhoCl in COS7 cells but not in neurons (data not shown). We then remade the library by amplifying the remaining PBPs after the sixth selection round and performing another selection.

We tested the results of this second selection by generating full-length and photocleaved PhoCl2c (see methods) and then pulling it down with PBP. Note that PhoCl2c is an improved version of the original PhoCl containing five mutations and with faster and more efficient cleavage (*Lu et al., 2021*). We found that PBP pulled down photocleaved PhoCl2c, but not full-length PhoCl2c (*Figure 4B*), suggesting that PhLIC could act as a photoactivatable system with low background. We further tested PhLIC by co-transfecting COS7 cells with Golgi-targeted PhoCl2c (GTS-PhoCl2c) and PBP-tagRFP and exposing them to 400 nm light. If PBP binds to photocleaved PhoCl2c but not full-length PhoCl2c, we would expect PBP-tagRFP to be expressed diffusely throughout the cell initially and then translocate to the Golgi after photocleavage of PhoCl2c (*Figure 4C*). Following expression in COS-7 cells, GTS-PhoCl2c localized in a concentrated oval in the center of the cell, consistent with Golgi localization. In contrast, PBP-tagRFP localized diffusely throughout the cytoplasm, with a ratio of labeling in the area colocalized with GTS-PhoCl2c vs. the rest of the cell of 1.5 ± 0.1 ($n$ = 11 cells, 2 experiments, *Figure 4D, E*). Following exposure to 400 nm light for 1 min and incubation in the dark for 3 hr, PBP-tagRFP became 3.3 ± 0.5 times more concentrated in the area colocalized with GTS-PhoCl2c staining than vs. the rest of the cell, a significant difference (p < 0.001, $n$ = 11, Wilcoxon, 2 experiments, *Figure 4D, F*). Note that exposure to ambient light did not affect the localization of PBP-tagRFP (*Figure 4—figure supplement 1B–E*), and exposing cells transfected only with tagRFP to 400 nm did not affect the localization of tagRFP (*Figure 4—figure supplement 1F–I*). These experiments suggest that PhLIC shows robust activation upon exposure to 400 nm light but no activation in ambient light.

## Photoactivatable GFE3

To generate a photoactivatable version of GFE3 using PhLIC (paGFE3), we fused PhoCl2c to transcriptionally controlled Gephyrin FingR (tagRFP-GPHN.FingR-PhoCl2c) and PBP to RING$_{XIAP}$ (PBP-E3). Note that tagRFP-GPHN.FingR-PhoCl2c performs two functions. It is targeted to inhibitory synapses, where it acts as an anchor that recruits PBP-E3 once PhoCl2c has been cleaved. It also acts as a label, allowing inhibitory synapses to be visualized before and after PhCl2c is cleaved (*Figure 5A*). To test whether paGFE3 labels inhibitory synapses, we expressed tagRFP-GPHN.FingR-PhoCl in cortical neurons in culture and co-labeled tagRFP and endogenous Gephyrin. The two labels are colocalized, consistent with tagRFP-GPHN.FingR-PhoCl labeling inhibitory synapses (*Figure 5—figure supplement 1B*). To test if paGFE3 caused background degradation of inhibitory synapses, we compared cultured cortical neurons transfected with paGFE3 (transcriptionally regulated tagRFP-GPHN.FingR-PhoCl and PBP-E3) with similar neurons transfected with tagRFP-GPHN.FingR-PhoCl alone and incubated for 5 days. Quantification of tagRFP labeling showed an 8% reduction, which was not significant ($n$ = 10 cells, 1 experiment, p > 0.4, Mann–Whitney, *Figure 5—figure supplement 1A*), consistent with no background degradation of Gephyrin in the dark. In contrast, when Cry2-tagRFP-GPHN.FingR and CIB1-RING$_{XIAP}$ were co-transfected into cultured cortical neurons and incubated for 5 days, tagRFP labeling was 48% less than when GPHN.FingR-Cry2-tagRFP was transfected by itself (650 ± 90 au, $n$ = 10 cells, 1 experiment vs. 1250 ± 110 au, $n$ = 9 cells, 1 experiment), a significant difference (p < 0.001, Mann–Whitney, *Figure 5—figure supplement 1A*).

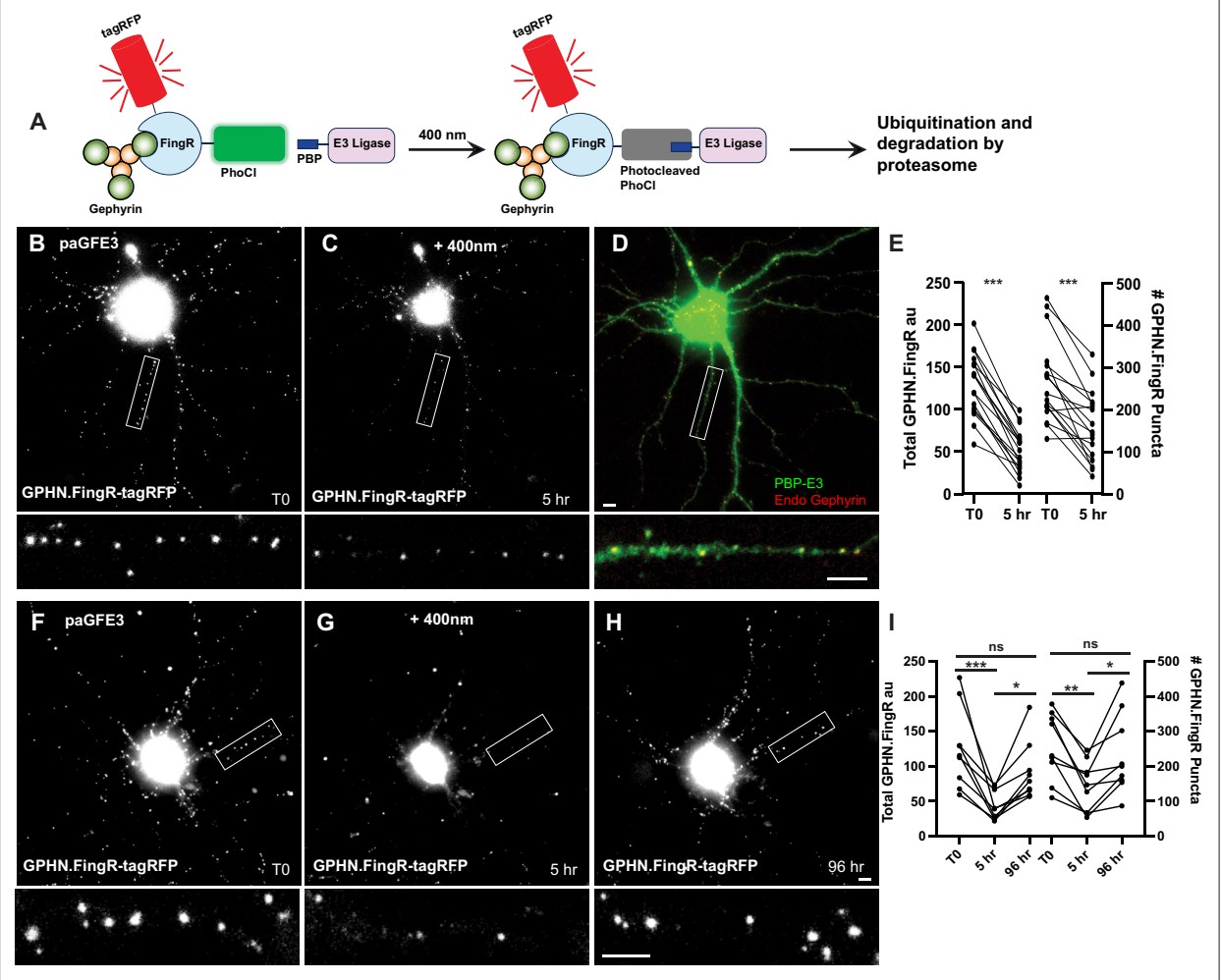

**Figure 5.** Reversible optogenetic ablation of inhibitory synapses with pa-GFE3. (**A**) Schematic of paGFE3. Initially, tagRFP-GPHN.FingR-PhoCl2c binds to Gephyrin and PBP-HA-E3 is unbound. After photocleavage with 400 nm light, PBP binds photocleaved PhoCl, recruiting the E3 ligase domain, which ubiquitinates Gephyrin and targets it for degradation by the proteasome. (**B**) Cultured cortical neuron expressing tagRFP-GPHN.FingR-PhoCl2c + PBP-HA-E3 (pa-GFE3) for 5 days. Closeup of the boxed area shown below. (**C**) Same neuron as in (B) 5 hr after illumination with 400 nm light for 1 min showing reduced labeling with tagRFP-GPHN.FingR-PhoCl2c. Closeup of the boxed area shown below. (**D**) Neuron in C immunostained for PBP-HA-E3 (green) and endogenous Gephyrin (red), showing sparse labeling for endogenous Gephyrin. Closeup of the boxed area shown below. (**E**) Quantification of the total amount of GPHN.FingR and number of GPHN.FingR puncta before and 5 hr after illumination with 400 nm light. ***p < 0.001, Wilcoxon. (**F**) Cultured cortical neuron expressing pa-GFE3. Closeup of the boxed area shown below. (**G**) Same neuron as in (F) 5 hr after illumination with 400 nm light for 1 min showing reduced labeling with tagRFP-GPHN.FingR-PhoCl2c. Closeup of the boxed area shown below. (**H**) Same neuron as in (**F**), (**G**) 4 days after illumination showing a recovery of tagRFP-GPHN.FingR-PhoCl2c labeling of synapses. Closeup of the boxed area shown below. (**I**) Quantification of the total amount of tagRFP-GPHN.FingR-PhoCl2c labeling and number of GPHN.FingR puncta showing recovery of synapses. ***p < 0.001, **p < 0.01, *p < 0.05, ns p > 0.1, Friedman multiple comparison test. Scale bars represent 5 µm.

The online version of this article includes the following source data and figure supplement(s) for figure 5:

**Source data 1.** Numerical source data for graphs in *Figure 5E, I*.

**Figure supplement 1.** pa-GFE3 has no background activity.

**Figure supplement 1—source data 1.** Numerical source data for graphs in *Figure 5—figure supplement 1A, E*.

To test whether the paGFE3 could mediate light-dependent degradation of Gephyrin, we expressed both components of paGFE3 (tagRFP-GPHN.FingR-PhoCl2c and PBP-E3) in cultured cortical neurons and incubated them for 5 days. Subsequently, neurons were intermittently exposed to 400 nm light for 1 min (see methods). After 5 hr, the cultures were reimaged. The neurons showed a 61 ± 3% reduction in tagRFP labeling (130 ± 9 vs. 52 ± 6 a.u., a significant difference, p < 0.001, Wilcoxon, *n* = 17 cells, 3 experiments, *Figure 5B–E*) and a 40 ± 6% drop in the number of GPHN.FingR puncta

(268 ± 24 vs. 161 ± 20 puncta), a significant change (p < 0.001, Wilcoxon, n = 17, 3 experiments, *Figure 5B–E*). Importantly, cells expressing transcriptionally regulated tagRFP-GPHN.FingR-PhoCl2c without PBP-RING had a much smaller reduction in labeling (10 ± 2%, *Figure 5—figure supplement 1C–E*) when exposed to 400 nm light as above, although it was statistically significant (p < 0.001, Wilcoxon, n = 14 cells, 3 experiments, *Figure 5—figure supplement 1*). However, the number of GPHN.FingR-labeled puncta did not show a significant change (0.4 ± 2%, p > 0.7, Wilcoxon, n = 14 cells, 3 experiments, *Figure 5—figure supplement 1*), indicating the reduction in total GPHN. FingR labeling is likely a result of photobleaching and not due to the decrease in Gephyrin. Thus, our results are consistent with cells expressing paGFE3 undergoing efficient degradation of Gephyrin puncta when exposed to 400 nm light.

To determine whether Gephyrin ablation mediated by paGFE3 is reversible, we transfected both components of pa-GFE3 in dissociated cortical neurons as before. Four days after transfection, cells were imaged and then exposed to 400 nm light for 1 min at time point T0. Five hours later, the cells were reimaged and incubated for another 4 days, at which point they were imaged again. We reasoned that the PhLIC-GFE3 complexes would be self-degrading, as was GFE3. Thus, in the absence of light, they should be degraded, and Gephyrin puncta should be allowed to be regenerated after a sufficient waiting period. Comparing the images, we found the total fluorescence from the tagRFP-GPHN.FingR-PhoCl2c label was reduced by 63 ± 5% between T0 and +5 hr, then increased by 130 ± 28% between +5 hr and +4 days, a significant difference (p < 0.001 for T0 to +5 hr; p < 0.05, +5 hr to +4 days, Friedman test, multiple comparisons, n = 9 cells, 2 experiments, *Figure 5F–I*). In addition, there was a 20 ± 9% decrease from T0 to +4 days, which was not significant (p > 0.1, Friedman test, multiple comparisons, *Figure 5I*), suggesting that paGFE3 is reversible. We found similar results when we counted the number of GPHN.FingR puncta which showed a 43 ± 7% reduction (p < 0.01, Friedman test, multiple comparisons) followed by a 76 ± 22% increase (p < 0.05, Friedman test, multiple comparisons), representing a nonsignificant 6 ± 12% decrease from the original (p > 0.99, Friedman test, multiple comparisons, n = 9 cells, 2 experiments). We also found that puncta before and after ablation were in similar locations (*Figure 5—figure supplement 1F*). Together, our results indicate that paGFE3 can mediate light-inducible degradation of endogenous Gephyrin that has virtually no background and is reversible.

## Chemically induced GFE3

We also generated a chemogenetic version of GFE3 (chGFE3) that can be used when an inducible version of GFE3 is needed, but paGFE3 is unsuitable. We used a chemically induced dimerization system based on *E. coli* dihydrofolate reductase (eDHFR) and the HaloTag protein. When combined with the small molecule TH, a fusion of trimethoprim (TMP) and HaloTag ligand, eDHFR and HaloTag form a bio-orthogonal complex in eukaryotic cells (*Ballister et al., 2014*). Furthermore, the formation of the complex is reversible because the binding of eDHFR to TMP is not covalent and can be outcompeted by adding excess TMP. We divided GFE3 into its two principal components, GPHN.FingR and RING$_{XIAP}$ and fused transcriptionally regulated GPHN.FingR to the HaloTag peptide and RING$_{XIAP}$ to eDHFR, to give tagRFP-GPHN.FingR-HaloTag and eDHFR-E3 (*Figure 6A*). As with tagRFP-GPHN. FingR-PhoCl2c, tagRFP-GPHN.FingR-HaloTag will both label inhibitory synapses and provide an anchor to which E3 can be recruited to initiate the degradation of Gephyrin. To test whether recruiting eDHFR-E3 to inhibitory synapses would lead to the loss of Gephyrin, we transfected cultured neurons with transcriptionally controlled tagRFP-GPHN.FingR-HaloTag and DHFR-E3 (chGFE3). After visualizing Gephyrin puncta, neurons were treated with 100 nM TH to induce dimerization. Incubation of the neurons with TH for 4 hr showed a reduction of Gephyrin by 73% ± 3% (p < 0.001, Wilcoxon, *Figure 6B–E*, n = 12 cells, 3 experiments) and a decrease in GPHN.FingR puncta by 53 ± 7% (p < 0.001, Wilcoxon, n = 12 cells, 3 experiments). Incubating the neurons with TH for 24 hr reduced total Gephryin labeling by 89 ± 2%, a significant reduction (p < 0.001, Wilcoxon, n = 12 cells, 3 experiments *Figure 6F–I*) and the number of GPHN.FingR puncta by 92 ± 0.1% (p < 0.001, Wilcoxon, n = 12 cells, 3 experiments). Immunocytochemistry against endogenous Gephyrin confirms the loss of inhibitory synapses at 4 and 24 hr (*Figure 6D, H*). To verify that Gephyrin loss depends on the dimerization between the GPHN.FingR and RING$_{XIAP}$ neurons were transfected with tagRFP-GPHN.FingR-HaloTag and SnapTag-E3 and treated with TH. There were nonsignificant increases in labeling with GPHN. FingR (+20 ± 16%, p > 0.5, Wilcoxon, n = 8, 2 experiments), and the number of GPHN.FingR puncta

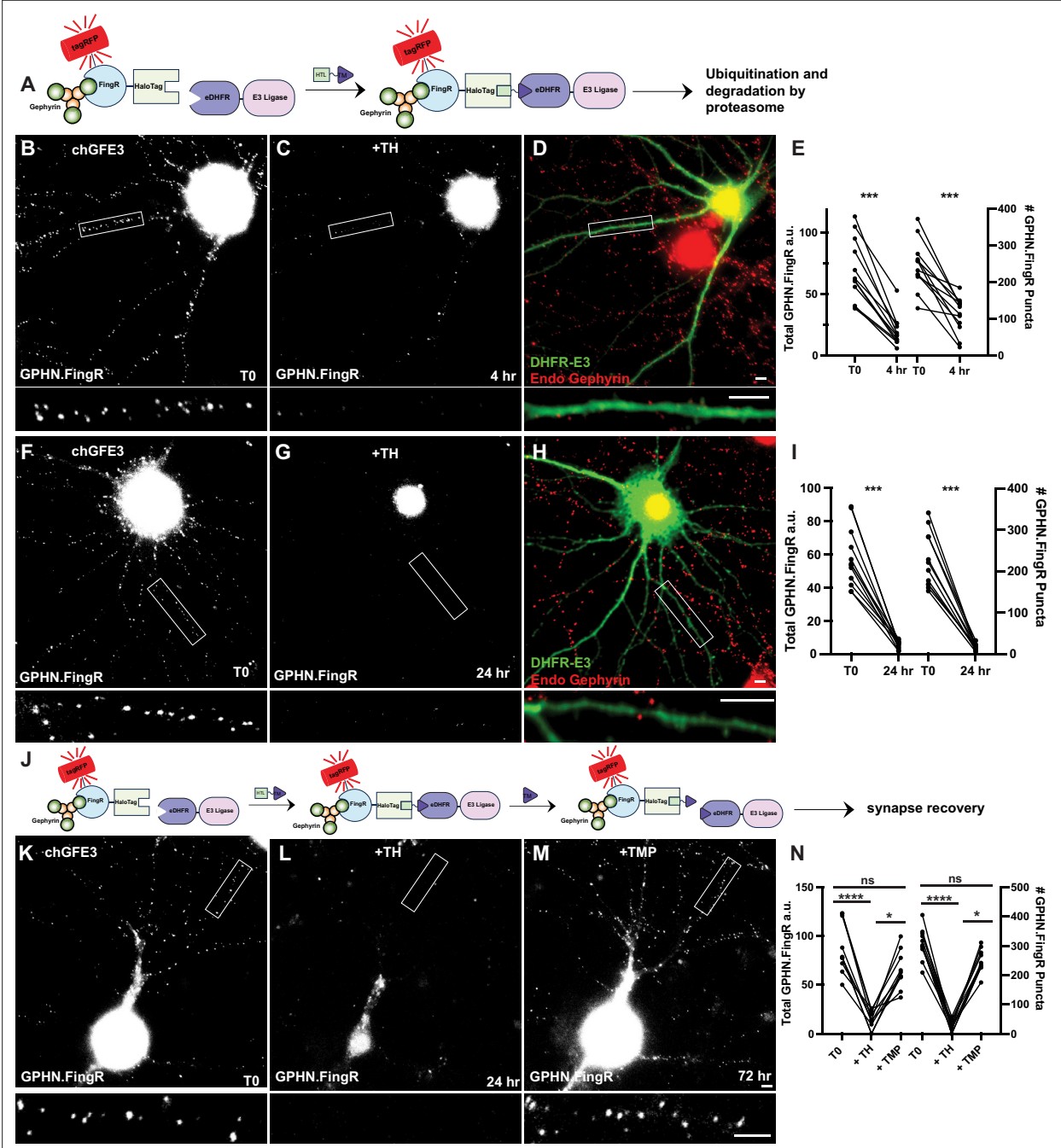

**Figure 6.** chGFE3 reversibly ablates Gephyrin. (**A**) Schematic of chGFE3. The addition of TMP-HaloTag ligand (TH) dimerizes HaloTag and eDHFR, leading to the recruitment of the E3 ligase to Gephyrin, which ubiquitinates and degrades it. (**B**) Cultured cortical neuron expressing GPHN.FingR-HaloTag and eDHFR-RING$_{XIAP}$ (chGFE3) for 4 days. Closeup of the boxed area shown below. (**C**) Same neuron as in (B) 4 hr after addition of 100 nM TH. Closeup of the boxed area shown below. (**D**) Immunocytochemistry of the neuron in (C) for endogenous Gephyrin (red) and eDHFR-E3$_{XIAP}$ (green). Note the lack of red puncta on dendrites labeled green. Closeup of the boxed area shown below. (**E**) Quantification of total Gephyrin labeled by GPHN.FingR-tagRFP and number of GPHN.FingR puncta after 4 hr of incubation with 100 nM TH. ***p < 0.001, Wilcoxon. (**F**) Cultured cortical neuron expressing GPHN.FingR-HaloTag and eDHFR-RING for 4 days. Closeup of the boxed area shown below. (**G**) Same neuron as in (**F**) after 24 hr incubation with 100 nm TH showing loss of Gephyrin. Closeup of the boxed area shown below. (**H**) Neuron in (**G**) immunostained for endogenous Gephyrin (red) and DHFR-E3 (green). Closeup of the boxed area shown below. (**I**) Quantification of the total amount of Gephyrin labeling by the GPHN.FingR after the addition of TH. ***p < 0.0001, Friedman multiple comparison test. (**J**) Schematic illustrating dissociation of chGFE3 with the addition of TMP. (**K**) Cultured cortical neuron expressing GPHN.FingR-HaloTag and eDHFR-RING$_{XIAP}$ for 4 days. Closeup of the boxed area shown below. (**L**) Same neuron as in (**K**) after 24 hr incubation with 100 nm TH showing loss of Gephyrin. Closeup of the boxed area shown below. (**M**) Same neuron as in (**L**) after 48 hr

*Figure 6 continued on next page*

*Figure 6 continued*

incubation with 100 µM TMP showing recovery of Gephyrin puncta. Closeup of the boxed area shown below. (**N**) Quantification of the total amount of Gephyrin labeling by the GPHN.FingR after the addition of TH and then TMP. ****p < 0.0001, *p < 0.05, ns p > 0.2 Friedman multiple comparison test. Scale bar represents 5 µm.

The online version of this article includes the following source data and figure supplement(s) for figure 6:

**Source data 1.** Numerical source data for graphs in *Figure 6E, I, N*.

**Figure supplement 1.** Chemogenetic ablation of inhibitory synapses with chGFE3.

**Figure supplement 1—source data 1.** Numerical source data for graphs in *Figure 6—figure supplement 1C*.

(+11 ± 10%, p > 0.99, Wilcoxon, *n* = 9, 2 experiments, *Figure 6—figure supplement 1A–C*). Endogenous Gephyrin was also unaffected in these cells (*Figure 6—figure supplement 1D*). Thus, our results are consistent with chGFE3 mediating efficient chemically induced degradation of Gephyrin.

The tagRFP-GPHN.FingR-Halotag/eDHFR-E3/TH complex should dissociate with the addition of excess TMP and reverse chGFE3 (*Figure 6J*). To test whether chGFE3 is reversible, we transfected cultured cortical neurons with GPHN.FingR-HaloTag and eDHFR-RING and treated with TH for 24 hr to induce degradation of Gephyrin, which led to a reduction of 83 ± 4% in GPHN.FingR-HaloTag labeling and 91 ± 2% reduction in the number of GPHN.FingR puncta (*Figure 6K, L, N*, p < 0.0001 for both, Friedman test, multiple comparisons, *n* = 10 cells, 2 experiments). We then added 100 µM TMP to the media for 24 hr to compete with TH to dissociate GPHN.FingR-RING$_{XIAP}$ dimers and block the formation of new dimers (*Figure 6M*). Imaging the neurons 48 hr after adding TMP later showed a 1450 ± 760% increase in the total GPHN.FingR labeling and 1168 ± 200% increase in Gephyrin puncta at 72 hr, which are significant (p < 0.05 for both, Friedman multiple comparison, *Figure 6M and N*). The amount of GPHN.FingR labeling at 72 hr compared to labeling at T0 was reduced by 24 ± 5% and the number of GPHN.FingR puncta decreased by 21 ± 4%, both of which were not significant (p > 0.2, Friedman multiple comparison, *Figure 6K, M, N*). Furthermore, immunocytochemistry shows colocalization between the GPHN.FingR and the GABA$_A$ receptor, the main ionotropic GABA receptor (*Goetz et al., 2007*), denoting a functional synapse (*Figure 6—figure supplement 1E*). Finally, we found that the Gephyrin puncta before and after ablation had similar distributions across dendrites (*Figure 6—figure supplement 1F*). Thus, our results suggest that in neurons expressing chGFE3, adding TH causes ablation of inhibitory neurons, and adding TMP reverses the effect.

## Discussion

In this study, we developed three new synapse ablators: PFE3, which ablates excitatory synapses; paGFE3, a photoactivatable version of the inhibitory synapse ablator; GFE3 and chGFE3, a chemically activated version of GFE3. All three tools ablate specific postsynaptic sites and can be expressed in particular cell types, allowing circuit breaking with twofold specificity.

PFE3 is composed of three distinct functional domains. The E3 ligase Mdm2 mediates the ubiquitination of PSD-95; the PIR domain recruits proteasomes to the synapse to degrade the ubiquitinated protein, and both domains are targeted to PSD-95 through PSD-95.FingR. Its design is based on work showing that coordinated actions of Mdm2 and PIR from PCDH 10 mediate the degradation of excitatory synapses (*Tsai et al., 2012*). These findings were corroborated when we found efficient degradation of PSD-95 and ablation of synapses was only achieved if all three components were included in PFE3. Expression of PFE3 caused highly efficient ablation of PSD-95 in cultured cortical neurons and blocked excitatory synaptic transmission in retinal ganglion cells in vivo.

Efficient elimination of excitatory synapses was achieved after 48 hr of PFE3 expression, which contrasts with PSD-95 knockout mice, where reductions in AMPA receptors and excitatory transmission were either not significant or small depending on the stage of development. One reason for this discrepancy could be that transcription of the MAGUK protein SAP-102, which is highly homologous to PSD-95, is upregulated in PSD-95 knockout mice and could provide compensation (*Coley and Gao, 2019*). PSD-95.FingR labels SAP-102[3], which suggests that PFE3 would degrade any SAP-102 that might otherwise compensate for the reduction in PSD-95. Thus, in the context of PFE3, the binding of PSD-95.FingR to MAGUKs homologous to PSD-95, such as SAP-102 and SAP97, could enable more effective and widespread ablation of excitatory synapses than if it bound only to PSD-95.

Alternatively, compensatory changes in transcription could be relatively slow, or they respond specifically to changes in mRNA levels rather than protein levels. In the future, it should be possible to use PFE3 to explore mechanisms of compensation that maintain homeostasis when protein levels change.

Another obvious application of PFE3 is to probe the role of PSD-95 in maintaining synaptic structure. For instance, what is the relationship between PSD-95 and dendritic spines? It would also be interesting to closely examine how synapses change before and after ablation of PSD-95. PSD-95 puncta grow back in cultured neurons, but what would happen in mature neurons in vivo? If they grow back, would they reconnect with their original presynaptic partners, and if so, would neural circuit function be changed following regrowth? It would also be interesting to see whether it is possible to permanently alter the connectivity of circuits after extended periods of PFE3 expression.

We added temporal and spatial control capability to GFE3 by enabling its activation with light (paGFE3) or chemicals (chGFE3). We used a standard method for regulating a protein where two distinct domains that are necessary and sufficient for protein function are fused to separate components of a light- or chemically induced protein complex. Previously, this approach has been used for controlling transcription (*Kennedy et al., 2010*), neurotransmitter release (*Liu et al., 2019*), gene editing (*Polstein and Gersbach, 2015*), recombination (*Duplus-Bottin et al., 2021*), and protease cleavage (*Sanchez and Ting, 2020*). However, background activation in photoactivatable complexes based on photoreceptors, such as cryptochrome 2 (*Kennedy et al., 2010*), phototropin (*Strickland et al., 2012*), or Vivid (*Kawano et al., 2015*), makes them unsuitable for regulating GFE3, which is highly potent. As an alternative, we developed a novel photoactivatable complex, PhLIC, based on the photocleavable protein PhoCl and the 10-mer PBP, which exhibited no measurable background when tested in the context of paGFE3.

This lack of background would make paGFE3 ideal for use in transgenic animals such as mice or zebrafish, where the complex would have to remain inert for long periods before activation. Expressing paGFE3 in a Gal4 (zebrafish) or Cre (mouse) driver line would allow inhibitory synapse ablation in specific cell types with precise temporal control without injecting viruses. In addition, with precise application of 400 nm light, it could be possible to ablate inhibitory synapses with single-cell spatial resolution. Furthermore, the PhLIC complex could be used to make photoactivatable versions of proteins other than GFE3. For instance, it could be used to create a split version of a potent toxin, which requires that no background activity be present until photoactivation causes it to kill or alter the function of specific cells at a particular time.

chGFE3 is based on a system similar to one for degrading exogenous proteins with eDHFR tags (*Etersque et al., 2023*). Our system has the same benefits of efficiency and reversibility, but it can target endogenous Gephyrin and ablate inhibitory synapses. In addition, we showed that, like paGFE3, chGFE3 has no background. Although its desirable qualities overlap with those of paGFE3, it could be used in contexts where it would be too difficult or time-consuming to photoactivate GFE3. For instance, if the target cells were spread over a relatively large region or when sustained ablation is necessary. One potential drawback to applying chGFE3 in vivo is the necessity of injecting the chemical dimerizer TH. However, it may be that TH can be administered systemically, particularly since both components of TH (the HaloTag ligand and TMP) can penetrate the brain when injected systemically (*Mohar et al., 2022*; *Ineichen et al., 2020*).

In conclusion, we have generated novel proteins for ablating excitatory and inhibitory synapses, which can be used to probe the structure and function of neural circuits.

# Materials and methods

## Key resources table

| Reagent type (species) or resource | Designation | Source or reference | Identifiers | Additional information |
|---|---|---|---|---|
| Recombinant DNA reagent | AAV-CAG-PFE3-P2A-GFP (plasmid) | This paper | RRID:Addgene_236178 | AAV version of constitutive PFE3 |
| Recombinant DNA reagent | AAV-TRE-PFE3-P2A-GFP (plasmid) | This paper | RRID:Addgene_236180 | AAV version of Tet-inducible PFE3 |

*Continued on next page*

*Continued*

| Reagent type (species) or resource | Designation | Source or reference | Identifiers | Additional information |
|---|---|---|---|---|
| Recombinant DNA reagent | AAV-CAG-RandE3-P2A-GFP | This paper | RRID:Addgene_236181 | AAV version of constitutive control plasmid |
| Recombinant DNA reagent | AAV-TRE-RandE3-P2A-GFP | This paper | RRID:Addgene_236179 | AAV version of inducible control plasmid |
| Recombinant DNA reagent | tagRFP-GPHN.FingR-PhoCl (plasmid) | This paper | RRID:Addgene_236183 | Half of photoactivatable GFE3 (paGFE3) |
| Recombinant DNA reagent | PBP-E3 (plasmid) | This paper | RRID:Addgene_236182 | Other half of paGFE3 |
| Recombinant DNA reagent | TRE-PFE3-HA | This paper | RRID:Addgene_236114 | Tet-inducible-PFE3-HA |
| Recombinant DNA reagent | tagRFP-GPHN.FingR-HaloTag (plasmid) | This paper | RRID:Addgene_236112 | Half of chemically inducible GFE3 (chGFE3) |
| Recombinant DNA reagent | eDHFR-E3 | This paper | RRID:Addgene_236113 | Other half of chGFE3 |
| Antibody | Anti-PSD-95 (mouse monoclonal) | Commercially available. | Novus Biologicals cat. # NB300-556 | 1:3000 |
| Antibody | Anti-GluA1 (rabbit polyclonal) | Commercially available. | Millipore Sigma cat. # ABN241 | 1:1000 |
| Antibody | Anti-GFP (chicken polyclonal) | Commercially available. | Fisher Scientific cat. # NC9510598 | 1:10,000 |
| Antibody | Anti-HA (rabbit polyclonal) | Commercially available. | Cell Signaling cat. # 3724 | 1:1000 |
| Antibody | Anti-MYC (chicken polyclonal) | Commercially available. | Novus Biologicals cat. #NB600-334 | 1:1000 |
| Genetic reagent (*M. musculus*) | CCK-Cre | Commercially available. | *Jackson Laboratories Cck*$^{tm1.1(cre)Zjh}$/J | Strain 012706 |
| Genetic reagent (*M. musculus*) | Ai27(RCL-hChR2(H134R)/tdT)-D | Commercially available. | Jackson Laboratories B6.Cg-*Gt(ROSA)26Sor*$^{tm27.1(CAG-COP4*H134R/tdTomato)Hze}$/J | Strain 012567 |

## Materials availability statement

All newly created plasmids are available through Addgene (see Key Resources Table).

## Preparation of cultured cortical neurons

Experimental protocols were conducted according to the National Institutes of Health guidelines for animal research and were approved by the Institutional Animal Care and Use Committee at the University of Southern California.

E19 wt Sprague Dawley rat embryos of both sexes were removed from the mother, and cortices were dissected in Hank's balanced salt solution (HBSS, Thermo Fisher Scientific, cat. # 14025076) supplemented with 0.1 mM HEPES (Thermo Fisher Scientific, cat. # 15630-080). Cortices were digested with neuronal isolation enzyme with Papain (Thermo Fisher Scientific, cat. # 88285) for 30 min at 37°C. The tissue was centrifuged at 1000 rpm for 1 min and washed three times with HBSS–HEPES. Subsequently, the tissue was triturated to dissociate the cells, and undissociated cells were separated out with a cell strainer (Falcon, cat. # 352340). Neurons were plated at a density of 8 × 10$^4$ on 2-well chambered cover glass (Cellvis, cat. # C2-1.5H-N) pre-treated overnight with 0.2 mg/ml poly-D-lysine (Sigma-Aldrich, cat. # P0899) and washed three times with water. Cells were plated in neurobasal media (NBM, Thermo Fisher Scientific, cat. # 21103-049) supplemented with 5 ml/l Glutamax (Thermo Fisher Scientific, cat. # 35050-061), 1 mg/l gentamicin solution (Thermo Fisher Scientific, cat. # 15750-078) 10 ml/l B27 (Thermo Fisher Scientific, cat. # 17504044), and 5% FBS (Hyclone, cat. # SH30071.03). Cells were cultured in a 5% CO$_2$ incubator at 37°C. Four hours after plating, the medium was diluted 1:3 with serum-free supplemented NBM. At 6 DIV, the medium was diluted 1:3 with supplemented NBM.

## Transfection, image capture, and analysis of cultured neurons

14–15 DIV cultured cortical neurons were transfected with a CalPhos mammalian transfection kit (Takara, cat. # 631312). Crystals were washed and replaced with 2:1 conditioned, fresh, supplemented NBM.

Live imaging of neurons was performed in imaging buffer consisting of HBSS (Thermo Fisher Scientific, cat. # 14025076) supplemented with 0.1 mM HEPES (Thermo Fisher Scientific, cat. # 15630-080). Imaging of immunostained and live cells was done on an Olympus IX81 inverted microscope with a ×60 water objective at ×1.0 zoom, an EM-CCD digital camera (Hamamatsu, cat. # C9100-02), GFP–mCherry and Cy5.5 filter cubes (Chroma Technology, cat. # 49000, cat. # 59022, and # SP105), an MS-2000 XYZ automated stage (Applied Scientific Instrumentation), an X-cite exacte mercury lamp (Excelitas Technologies) and Metamorph software (Molecular Devices).

Image analyses were performed using ImageJ software. SynQuant (*Wang et al., 2020*) was used to detect PSD-95.FingR puncta for excitatory synapses and GPHN.FingR puncta for inhibitory synapses. Synapse detection by SynQuant was reviewed, and false positive signals were manually removed. The total amount of PSD-95 or Gephyrin FingR was calculated by taking the sum of the product of the area and intensity for each punctum.

## Immunocytochemistry of cultured neurons

Cells were fixed with 4% PFA (Electron Microscopy Sciences, cat. # 15714) for 5 min and washed three times for 5 min with PBS. Cells were then permeabilized and blocked with blocking buffer (1% bovine serum albumin [BSA], 5% normal goat serum, and 0.1% Triton X-100 in PBS) for 30 min. The primary antibody was then diluted in blocking buffer and added to the cells for 1 hr. After three 5 min washes with PBS, the secondary antibody was diluted in blocking buffer and added to cells for 1 hr in the dark. The cells were again washed three times with PBS and imaged. Primary antibody concentrations used: mouse anti-PSD-95 (1:3000, Novus Biological, cat. # NB300-556), rabbit anti-GluA1 (1:1000, Millipore Sigma, ABN241), and chicken anti-GFP (1:10,000, Aves Labs, cat. # NC9510598), Rabbit anti-HA (1:1000, Cell Signaling, cat. # 3724), chicken anti myc (1:1000, Novus Biologicals cat. # NB600-334). Secondary antibodies used were the following from Thermo Fisher Scientific at 1:1000: goat anti chicken-Alexa Fluor 488 (cat. # A-11039), goat anti-rabbit Alexa Fluor 594 (cat. # A-11012), goat anti-rabbit Alexa Fluor 647 (cat. # A-21245), and goat anti-mouse Alexa Fluor 594 (cat. # A-11032).

## mRNA display

mRNA display was carried out as described in *Gross et al., 2013* with the following modifications. The mRNA display library was constructed with a 5′ constant region including a T7 promoter, transcription start sequence, and a ΔTMV translation enhancer region followed by a semi-random variable region encoding PBP followed by a 3′ constant region containing a short flexible linker, HA tag and the splint sequence used to ligate the transcript to puromycin (Keck oligonucleotide facility, Yale). The cDNA library was purified with Urea-PAGE, and Klenow extension (NEB, cat. # 0210S) was used to generate the dsDNA library. The library was then transcribed with T7 RNA Polymerase and ligated to puromycin (pF30P, Oligo Synthesis at Yale School of Medicine) using T4 DNA Ligase (NEB). The mRNA–puromycin fusion was purified with Urea-PAGE and electroeluted with an Elutrap (Schleicher & Schuell BioScience). The library was translated with rabbit reticulocyte lysate (Promega cat. # L4960) and purified with dT-25 BIOTEG beads (NEB cat. # S1419), eluted in water, and desalted with Centrisep columns (Princeton Separations cat. # CS-901). The purified mRNA–peptide fusion was reverse transcribed with Superscript IV (Invitrogen cat. # 18091050).

The purified library of peptide–cDNA conjugates was then incubated with biotinylated, photocleaved PhoCl and immobilized on streptavidin or neutravidin beads (Thermo Fisher Scientific, cat. # 29200, 20353) in selection buffer (20 mM Tris-HCl, pH 8.0, 150 mM NaCl, 0.02% Tween-20, 0.5 mg/ml BSA, 1 mM DTT, 1% FBS, and 0.2 mM D-Biotin). Full-length PhoCl was added to the reaction in excess. Incubation of the library with cleaved PhoCl for rounds 1 and 2 was conducted at 4°C, rounds 3–5 at room temperature, and round 6 at 30°C. The beads were then washed three times with selection buffer and once with TBST (20 mM Tris-HCl, pH 8.0, 150 mM NaCl, 0.02% Tween-20). The beads containing cleaved PhoCl and any peptides bound were used directly as templates for PCR amplification of the library. The resulting PCR product was used as the library for the subsequent selection round. The library underwent six rounds of selection.

## Ubiquitination and proteasome dependency of PFE3

COS-7 cells (ATCC cat. # CRL-1651) between P5 and P10 were cultured in Dulbecco's modified Eagle's medium (DMEM) (ATCC, cat. # 30-2002) supplemented with 10% FBS (Hyclone, cat. # SH30071.03) and 12.5 mg/l gentamicin solution (Thermo Fisher Scientific, cat. # 15750-078) in a 5% $CO_2$ incubator at 37°C. Cells were grown to 50% confluency and co-transfected with PSD-95.myc, reverse tetracy-cline transactivator (rtTA) and doxycycline-inducible PSD-95.FingR-Mdm2.RING or Rand.FingR-Mdm2. RING using Lipofectamine 2000 transfection reagent (Thermo Fisher Scientific, cat. # 11668019). Cells were found to be negative for mycoplasma using an InvivoGen test. Cell type was confirmed through morphological examination.

Twenty-four hours after transfection, cells were treated with 1 µg/ml doxycycline (Sigma-Aldrich, cat. # 5207) to induce expression of PSD-95.FingR-Mdm2.RING and 20 µM TAK243 (MedChemEx-pres, cat. # HY-100487) to block ubiquitination for 4 hr. Cells were then rinsed with PBS and scraped in Lysis Buffer (150 mM NaCl, 1 mM EDTA, 20 mM Tris, pH 8.0, 1% NP-40) with freshly added complete mini protease inhibitor cocktail (Roche, cat. # 04693124001). After incubating on ice for 30 min, the total cell lysate was centrifuged at 10,000 rpm for 10 min at 4°C to pellet the insoluble fraction of the lysate. The concentration of the lysate was determined with a Bradford Protein Assay (Bio-Rad, cat. # 5000201). 30 µg of cell lysate was denatured in Laemmli Buffer (Bio-Rad, cat. # 1610747) with 10% 2-mercaptoethanol and ran on precast Any kD SDS–PAGE protein gels (Bio-Rad, cat. # 4569034). Proteins were then transferred onto low-fluorescence PVDF membrane (Bio-Rad, cat. # 1620264) and probed with chicken anti-myc (1:3000, Novus Biologicals NB600-334) and mouse anti- α-tu-bulin (1:5000, Sigma T6199). Secondary antibodies were goat anti-chicken Alexa Fluor 680 (1:1000, Abcam, cat. # ab175779) and goat anti-mouse Alexa Fluor 750 (1:1000, Thermo Fisher Scientific, cat. # A-21037).

Blots were imaged with the Odyssey Infrared Imaging System (LI-COR Biosciences), and protein levels were determined using ImageJ Software (US National Institutes of Health). PSD-95 levels to α-tubulin levels.

## COS7 cell culture and transfection

COS-7 cells (ATCC cat# CRL-1651) were cultured in DMEM (ATCC, cat. # 30-2002) supplemented with 10% FBS (Hyclone, cat. # SH30071.03) and 12.5 mg/l gentamicin solution (Thermo Fisher Scientific, cat. # 15750-078) in a 5% $CO_2$ incubator at 37°C. Cells were grown to 50% confluence on coverslips (VWR, cat. # 48393-059) and co-transfected Golgi-targeted PhoCl and tagRFP or PBP-tagRFP using Lipofectamine 2000 transfection reagent (Thermo Fisher Scientific, cat. # 11668019). Twenty-four hours after transfections, cells were transferred to imaging buffer.

## Excitatory synapse ablation with PFE3

Neurons for PSD-95 ablation were transfected with 1 µg pCAG:Zreg:tagRFP-PSD-95.FingR, 500 ng CMV:Tet3G, and 1 µg TRE-PFE3-HA or 1 µg TRE-PFE3-P2A-GFP as described above and incubated for 4 days. Neurons were then transferred to the imaging buffer for imaging, and the positions of each neuron were saved in the Metamorph Software. Neurons were then transferred back into the cultured media, and doxycycline was added to 1 µg/ml. After incubation for 48 hr, the same neurons were imaged to track changes in PSD-95.FingR. To confirm the loss of endogenous PSD-95, neurons were immediately fixed and immunostained after imaging. To recover PSD-95 puncta, neurons were washed with conditioned media five times to remove the doxycycline and incubated in conditioned media for 5 days.

## Photocleavage of PhoCl in live cells

Neurons for synapse ablation were transfected with 1 µg pCAG:tagRFP-GPHN.FingR-PhoCl2c and 500 ng pCAG:PBP-E3 and incubated for 5 days. COS7 cells were transfected with 200 ng pGW:GTS-PhoCl2c and 50 ng pCAG:PBP-tagRFP and incubated overnight. The COS7 cells and neurons were then imaged in imaging buffer, and positions for each cell were recorded and saved. Each cell was subsequently illuminated with ~400 nm light at 40 mW/cm$^2$ for 10 s every 30 s for a total of 3 min (total 1 min illumination time). The cells were then transferred to media and incubated at 37°C. Following the incubation, cells were again transferred to imaging buffer and imaged.

## PBP binding to cleaved PhoCl

PhoCl2c was expressed in HEK293T cells (ATCC cat. # CRL-3216) cultured in DMEM (ATCC, cat. # 30-2002) supplemented with 10% FBS (Hyclone, cat. # SH30071.03), 125 mg/l gentamicin solution (Thermo Fisher Scientific, cat. # 15750-078) and GlutaMAX Supplement (Thermo Fisher Scientific, cat. # 35050061) in a 5% $CO_2$ incubator at 37°C. Cells were grown to 75% confluency in 150 mm cell culture dishes (Corning, cat. # 430599) and transfected using PEI (Polysciences, cat. # 24765-1) with Myc-PhoCl-GST or Myc-PhoCl-HRV3C.cleavage site-GST. After 72 hr of expression, cells were rinsed with PBS and lysed with ice-cold Lysis Buffer (25 mM Tris-HCl pH 7.4, 150 mM NaCl, 1 mM EDTA, 1% NP-40, 5% glycerol) supplemented with freshly added complete mini protease inhibitor cocktail (Roche, cat. # 04693124001). The total cell lysate was centrifuged at 5000 rpm for 30 min at 4°C, and the supernatant was bound to Glutathione Agarose beads (Thermo Fisher Scientific, cat. # 16100) at 4°C overnight. The beads were then washed five times with Wash Buffer 5 (25 mM Tris-HCl pH 7.4, 150 mM NaCl, 1 mM EDTA, 5% glycerol). To generate photocleaved PhoCl2c, beads with Myc-PhoCl-GST were transferred to glass chambers (Cellvis, cat. # C2-1.5H-N) and exposed to 400 nm LED 100 mW/cm² for 20 s every minute for a total of 5 min. The beads were then transferred to an ultracentrifuge tube and rotated at 4°C overnight in Lysis Buffer to allow the dissociation of PhoCl2c. To generate full-length PhoCl2c, beads containing Myc-PhoCl2c-HRV3C.cleavage site-GST were incubated with HRV3C protease (Thermo Fisher Scientific, cat. # 88947) at 4°C overnight. The supernatant containing purified photocleaved or full-length PhoCl2c was separated from the beads using Spin-X centrifuge tube filters (Corning, cat. # CLS8162). Photocleavage and HRV3C cleavage of photocleaved and full-length PhoCl2c were confirmed with SDS–PAGE.

## Electrophysiology

Experimental protocols were conducted according to the National Institutes of Health guidelines for animal research and were approved by the Institutional Animal Care and Use Committee at the University of California, Berkeley. To express Channelrhodopsin-2 in type 6 cone bipolar cells, we crossed a CCK-Cre mouse line (Jackson Laboratory strain 012706) with the Jackson Laboratory Ai27 mouse, which contains a floxed Channelrhodopsin2-TdTomato fusion protein sequence. Mice were of both sexes and 4–6 months of age. Mice were intravitreally injected with either the PFE3 or control virus (1.5 µl). After 3–4 weeks, retinas were isolated and mounted in a recording chamber for patch clamp recording, as described previously (*Jones et al., 2012*). Retinas were perfused in ACSF bubbled with 95% $O_2$/5% $CO_2$. ACET (1 µM) and L-AP4 (10 µM) were added to block synaptic transmission from photoreceptors to Off and On bipolar cells, respectively. Infected cells were identified by GFP expression. GFP-positive On α-RGCs were identified by their large cell bodies, and their identity was confirmed by dye filling. Full-field 495 nm light (0.9 mW/cm²) was delivered to the retina for 10ms to trigger EPSCs. Cells were voltage-clamped at −−60 mV.

## Chemogenetic GFE3

Cultured cortical neurons were prepared and transfected with 1 µg tagRFP-GPHN.FingR-HaloTag and 250 ng eDHFR-E3 as described above at 14 DIV. Neurons were imaged at 18 DIV in imaging buffer, and the coordinates of each neuron were saved in the Metamorph Software. Immediately after imaging, neurons were transferred back to the culture media, and TH was added to the medium at 100 nM final concentration. After 4 or 24 hr of incubation, neurons were transferred to the imaging buffer, and the same neurons were imaged to track changes in GPHN.FingR. To confirm the loss of endogenous Gephyrin, neurons were immediately fixed and immunostained, as described above. To recover inhibitory synapses, the cultured neurons were washed three times with conditioned media and incubated with conditioned media with 50 mM TMP for 48 hr.

## Statistical analysis

We used nonparametric tests for paired (Wilcoxon), unpaired (Mann–Whitney), or multiple (>2) samples (Friedman multiple comparison) because we could not be certain that the data being tested were normally distributed.

## Acknowledgements

We thank all Arnold lab members for helpful discussions. We also thank Alexandra Delgadillo, Jacqueline Rivera, and Ben Shapero for their technical assistance. We thank Scott Nawy from the Kramer lab for compiling the data in *Figure 3*. This work was supported by a grant (NS115610) to DA from NINDS and the Brain Initiative. mRNA display was originally developed by Richard Roberts, who helped establish it in the Arnold lab.

## Additional information

### Funding

| Funder | Grant reference number | Author |
| --- | --- | --- |
| National Institute of Neurological Disorders and Stroke | NS115610 | Don B Arnold |

The funders had no role in study design, data collection, and interpretation, or the decision to submit the work for publication.

### Author contributions

Aida Bareghamyan, Formal analysis, Investigation, Visualization, Writing – original draft, Writing – review and editing; Changfeng Deng, Xiaocen Lu, Wei Zhang, Robert E Campbell, David M Chenoweth, Resources; Sarah Daoudi, Shubhash C Yadav, Investigation; Richard H Kramer, Supervision; Don B Arnold, Conceptualization, Resources, Supervision, Funding acquisition, Methodology, Writing – original draft, Project administration, Writing – review and editing

### Author ORCIDs

Richard H Kramer (iD) https://orcid.org/0000-0002-8755-9389
Don B Arnold (iD) https://orcid.org/0000-0001-7378-1440

### Ethics

This study was performed in strict accordance with the recommendations in the Guide for the Care and Use of Laboratory Animals of the National Institutes of Health. All of the animals were handled according to the Institutional Animal Care and Use Committee (IACUC) approved protocols 21142 of the University of Southern California or AUP-2016-04-8700-3 of University of California, Berkeley.

Reviewer #1 (Public review): https://doi.org/10.7554/eLife.103757.3.sa1
Reviewer #2 (Public review): https://doi.org/10.7554/eLife.103757.3.sa2
Author response https://doi.org/10.7554/eLife.103757.3.sa3

## Additional files

### Supplementary files

MDAR checklist

### Data availability

Figures 1–6—source data contain the numerical data used to generate Figures 1–6.

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
