## [Editor Report · eLife Assessment]

This **compelling** study introduces a set of novel genetically encoded tools for the selective and reversible ablation of excitatory and inhibitory synapses. These new tools enable selective and efficient ablation of excitatory synapses, and photoactivatable and chemically inducible methods for inhibitory synapse ablation in specific cell types, providing **valuable** methods for disrupting neural circuits. This approach holds broad potential for investigating the roles of specific synaptic input onto genetically determined cells.

---

## [Referee Report · Reviewer #1 (Public review)]

Summary:

This work is a continuation of a previous paper from the Arnold group, where they engineered GFE3, which allows to specifically ablate inhibitory synapses. Here, the authors generate 3 different actuators:

(1) An excitatory synapse ablator.

(2) A photoactivatable inhibitory synapse ablator.

(3) A chemically inhibitory synapse ablator.

Following initial engineering, the authors present characterization and optimization data to showcase that these new tools allow one to specifically ablate synapses, without toxicity and with specificity. Furthermore, they showcase that these manipulations are reversible.

Altogether, these new tools would be important for the neuroscience community.

Strengths:

The authors convincingly demonstrate the engineering, optimization and characterization of these new probes. The main novelty here is the new excitatory synapse ablator, which has not been shown yet and thus could be a valuable tool for neuroscientists.

Weaknesses:

The authors have convincingly demonstrated the use of these tools in cultured neurons. The biggest weakness is the limited information given for the use of these tools for in vivo studies. The authors provide one example of the use of these new tool to study retinal circuits, and show evidence that the excitatory synapse ablator reduces synaptic transmission in retinal slices. Still, more work will be required to use this tool in intact neuronal circuits. It remains unclear if it would be trivial to characterize how well these tools express and operate in vivo. This could be substantially different and present some limitations as to the utility of these tools.

---

## [Referee Report · Reviewer #2 (Public review)]

Summary:

This study introduces a set of genetically encoded tools for the selective and reversible ablation of excitatory and inhibitory synapses. Previously, the authors developed GFE3, a tool that efficiently ablates inhibitory synapses by targeting an E3 ligase to the inhibitory scaffolding protein Gephyrin via GPHN.FingR, a recombinant, antibody-like protein (Gross et al., 2016). Building on this work, they now present three new ablation tools: PFE3, which targets excitatory synapses, and two new versions of GFE3-paGFE3 and chGFE3-that are photoactivatable and chemically inducible, respectively. These tools enable selective and efficient synapse ablation in specific cell types, providing valuable methods for disrupting neural circuits. This approach holds broad potential for investigating the roles of specific synaptic input onto genetically determined cells.

Strengths:

The primary strength of this study lies in the rational design and robust validation of each tool's effectiveness, building on previous work by the authors' group (Gross et al., 2016). Each tool serves distinct research needs: PFE3 enables efficient degradation of PSD-95 at excitatory synapses, while paGFE3 and chGFE3 allow for targeted degradation of Gephyrin, offering spatiotemporal control over inhibitory synapses via light or chemical activation. These tools are efficiently validated through robust experiments demonstrating reductions in synaptic markers (PSD-95 and Gephyrin) and confirming reversibility, which is crucial for transient ablation. By providing tools with both optogenetic and chemical control options, this study broadens the applicability of synapse manipulation across varied experimental conditions, enhancing the utility of E3 ligase-based approaches for synapse ablation.

Weaknesses:

While this study provides valuable tools and addresses many critical points for varidation, examining potential issues with specificity and background ubiquitination in further detail could strengthen the paper. For PFE3, the study demonstrates reductions in both PSD-95 and GluA1. In their previous work, GFE3 selectively reduced Gephyrin without affecting major Gephyrin interactors or other PSD proteins. Clarifying whether PFE3 affects additional PSD proteins beyond GluA1 would be important for accurately interpreting results in experiments using PFE3. Additionally, further insight into PFE3's impact on inhibitory synapses would be valuable to assess the excitatory specificity and potential for circuit-level applications. For paGFE3 and chGFE3, the E3 ligase (RING domain of Mdm2) is overexpressed and thus freely diffusible within the cell as a separate construct. Although the authors show that Gephyrin is not significantly reduced without light or chemical activation, it remains possible that other proteins, particularly non-synaptic proteins, could be ubiquitinated due to the presence of freely diffusing E3 ligase in cytosol. Addressing these points would clarify the strengths and limitations of tools, providing users with valuable information.

---

## [Author Response]

The following is the authors’ response to the original reviews.

**Public Reviews**

**Reviewer #1:**
The biggest concern in this regard is: that almost all the characterization is performed in cultured dissociated neurons…

While it is true that most of the characterization done in this paper was in cultured neurons, we verified that PFE3 mediates functional ablation of excitatory synapses in vivo (Fig. 3). Furthermore, the GPHN.FingR-XIAP (GFE3), a protein very similar to the complex formed following activation of paGFE3 and chGFE3, has been extensively tested by us and others in vivo(1-4).

**Reviewer #2:**
For paGFE3 and chGFE3, the E3 ligase (RING domain of Mdm2) is overexpressed throughout cells as a separate construct. Although the authors show that Gephyrin is not significantly reduced without light or chemical activation, it remains possible that other proteins could be ubiquitinated due to the overexpressed E3 domain.

In our previous paper(1), we tested neurons under 3 conditions: 1. expressing a construct similar to PBP-E3, consisting of a FingR with a randomized binding domain fused to the same XIAP ring domain used in paGFE3 and chGFE3 (RAND-E3). 2. expressing GPHN.FingR. 3. not expressing any exogenous proteins (control neurons). In each case, we found that expression of a variety of excitatory and inhibitory synaptic proteins was not significantly different when exposed to either of these exogenous proteins compared with control neurons.

**Recommendations for the authors:**
(1) Can the authors use the tools to show the ablation of endogenous PSD95 without FingR overexpression?

The experiments described in Fig. 3 are an example of this type of experiment. Furthermore, the PSD-95.FingR was extensively tested and has been used in dozens of studies without any indication that its expression alters cellular function or morphology. Note also that the transcriptional regulation system of PSD-95.FingR limits the expression such that there is virtually no background, so it is not really being overexpressed.

(2) I am missing some control experiments for the excitatory synapses ablator- can the authors show that cells transfected with the plasmid and no DOX, show similar numbers of synapses as neurons without transfection?

We have added an experiment comparing cells expressing PSD-95.FingR alone, and others expressing PFE3 with no Dox. We found that the two types of cells express amounts of PSD-95 that are not significantly different (Fig. S2L).

(3) I am not quite sure how they used paired statistics on staining since they could only stain the cell at the end of the experiment. Are the comparisons performed on different cells?

These experiments were done on the same cells. However, the methods of labeling were different- the initial counting of synapses was done, so we agree with the reviewer that it would be best not to use a paired analysis. Accordingly, we have changed Figs. 1F and 2D.